# The Role of Chitosan as a Possible Agent for Enteric Methane Mitigation in Ruminants

**DOI:** 10.3390/ani9110942

**Published:** 2019-11-09

**Authors:** Rafael Jiménez-Ocampo, Sara Valencia-Salazar, Carmen Elisa Pinzón-Díaz, Esperanza Herrera-Torres, Carlos Fernando Aguilar-Pérez, Jacobo Arango, Juan Carlos Ku-Vera

**Affiliations:** 1Faculty of Veterinary Medicine and Animal Science, University of Yucatan, Carretera Merida-Xmatkuil km 15.5. Apdo. 4-116 Itzimna, C.P. 97100 Merida, Yucatan, Mexico; rafax77@hotmail.com (R.J.-O.); caperez@correo.uady.mx (C.F.A.-P.); 2College of the Southern Border (ECOSUR), Livestock and Environment, Carretera Panamericana—Periferico Sur, C.P. 29290 San Cristobal de las Casas, Chiapas, Mexico; saraudea@gmail.com; 3Faculty of Veterinary Medicine and Animal Science, Juarez University of Durango, Carr Durango—Mezquital km 11.5, C.P. 34307 Durango, Mexico; carmencita84@hotmail.com (C.E.P.-D.); hetoes99@yahoo.com.mx (E.H.-T.); 4International Center for Tropical Agriculture (CIAT), km 17, Recta Cali-Palmira, Palmira C.P. 763537 Valle del Cauca, Colombia; j.arango@cgiar.org

**Keywords:** ruminant, chitosan, fermentation pattern, propionic acid, methane

## Abstract

**Simple Summary:**

Ruminant husbandry is one the largest contributors to greenhouse gas emissions from the agriculture sector, particularly of methane gas, which is a byproduct of the anaerobic fermentation of structural and non-structural carbohydrates in the rumen. Increasing the efficiency of production systems and decreasing its environmental burden is a global commitment, thus methane mitigation is a strategy in which to reach these goals by rechanneling metabolic hydrogen (H_2_) into volatile fatty acids (VFA) to reduce the loss of energy as methane in the rumen, which ranges from 2% (grain rations) to 12% (poor-quality forage rations) of gross energy intake. A strategy to achieve that goal may be through the manipulation of rumen fermentation with natural compounds such as chitosan. In this review, we describe the effects of chitosan on feed intake and rumen fermentation, and present some results on methanogenesis. The main compounds with antimethanogenic properties are the secondary metabolites, which are generally classified into five main groups: saponins, tannins, essential oils, organosulfurized compounds, and flavonoids. Novel compounds of interest include chitosan obtained by the deacetylation of chitin, with beneficial properties such as biocompatibility, biodegradability, non-toxicity, and chelation of metal ions. This compound has shown its potential to modify the rumen microbiome, improve nitrogen (N) metabolism, and mitigate enteric methane (CH_4_) under some circumstances. Further evaluations in vivo are necessary at different doses in ruminant species as well as the economic evaluation of its incorporation in practical rations.

**Abstract:**

Livestock production is a main source of anthropogenic greenhouse gases (GHG). The main gases are CH_4_ with a global warming potential (GWP) 25 times and nitrous oxide (N_2_O) with a GWP 298 times, that of carbon dioxide (CO_2_) arising from enteric fermentation or from manure management, respectively. In fact, CH_4_ is the second most important GHG emitted globally. This current scenario has increased the concerns about global warming and encouraged the development of intensive research on different natural compounds to be used as feed additives in ruminant rations and modify the rumen ecosystem, fermentation pattern, and mitigate enteric CH_4_. The compounds most studied are the secondary metabolites of plants, which include a vast array of chemical substances like polyphenols and saponins that are present in plant tissues of different species, but the results are not consistent, and the extraction cost has constrained their utilization in practical animal feeding. Other new compounds of interest include polysaccharide biopolymers such as chitosan, mainly obtained as a marine co-product. As with other compounds, the effect of chitosan on the rumen microbial population depends on the source, purity, dose, process of extraction, and storage. In addition, it is important to identify compounds without adverse effects on rumen fermentation. The present review is aimed at providing information about chitosan for dietary manipulation to be considered for future studies to mitigate enteric methane and reduce the environmental impact of GHGs arising from livestock production systems. Chitosan is a promising agent with methane mitigating effects, but further research is required with in vivo models to establish effective daily doses without any detrimental effect to the animal and consider its addition in practical rations as well as the economic cost of methane mitigation.

## 1. Introduction

Mitigation of enteric methane production in ruminants results in two main advantages: the first is that CH_4_ is a short-lived climate forcer that remains for only 12.2 years in the atmosphere when compared with CO_2_, which remains for decades [1,2]. The second advantage is that methane mitigation will increase the efficiency of production in livestock systems, particularly those that include ruminants [3]. Conditions within the rumen are favorable for the hydrogenotrophic methanogens (*Methanobrevibacter*, *Methanomicrobium*, *Methanosphaera, Methanosarcina, Methanobacterium,* and rumen cluster *C*) [4] that reduce CO_2_ with the metabolic H_2_ produced (87–90% in the rumen) during anaerobic fermentation of glucose, which results in the synthesis of CH_4_ [5,6] and the remaining 10 to 13% that is generated in the hindgut [7]. Methane is eructated in large amounts by all ruminant species and has a gross energy content of 55.65 MJ per kilogram [8]. Emissions of enteric CH_4_ vary according to dry matter intake, growth rate, housing, live weight, level of production, ration composition, and rumen fermentation pattern [9,10]. Ration and type of feed affect the availability of metabolic hydrogen for CH_4_ synthesis in the rumen; feeding concentrate and grains (low fiber diets) reduce the population of methanogens (range 10^7^ to 10^9^ g^−1^) while more propionic acid is synthesized, which utilizes H_2_, whereas pasture-fed (methanogens 10^9^ to 10^10^ g^−1^) ruminants yield higher concentrations of acetic acid in rumen liquor, resulting in a higher availability of metabolic H_2_ [11]. However, high-grain levels in rations can decrease rumen pH and result in health problems in ruminants such as acidosis [5,10,11]. During the last decade, methane mitigation strategies have been the subject of intensive study by several research groups [12]. Strategies for CH_4_ mitigation that have been evaluated are varied and include the inoculation of exogenous bacterial strains [13], vaccine development, biological control, prebiotics, probiotics [6], defaunation (protozoa or methanogens), and the identification of natural compounds in plants used as feed additives [14]. These are not always applicable in practice and the results of some strategies have shown some variability [15], with variation attributed to the concentration of the compound of interest in the foliage, pods, or extracts, interaction or agonistic compounds, growth stage, climate conditions, manipulation compounds, in vitro or in vivo experiments, daily intake, and bioavailability [16,17]. Dietary manipulation is of relevance because the diet of animals has a great influence on the production of methane [3,10,18]; therefore, in the last decade, more intensive research on natural compounds for livestock has increased with the objective of reducing rumen CH_4_ without affecting rumen fermentation and energy utilization mainly in in vitro trials [19,20]. 

Plants produce a wide variety of bioactive compounds and secondary metabolites [21]. Some of them have been shown to be useful for manipulating some metabolic processes in ruminants and selectively modulating microbial populations in the rumen, allowing for an improvement in fermentation, nitrogen metabolism, and in reducing methane production [22]. There is evidence that certain natural compounds have the potential to mitigate methane production in different species of ruminants and improve their productive functions (meat or milk), even with the use of different diets [23]. A range of feed additives including antibiotics has achieved a reduction of methane, but the regulatory law prohibits the use of these compounds (European Union Regulation 1831/2003/EC 2006) in some countries because of the lack of social acceptance, residues in food products, and resistant strains of pathogens, which has increased the necessity of finding natural compounds [6,24,25]. There has been growing interest in utilizing natural compounds as a novel strategy to mitigate CH_4_ by selectively targeting rumen methanogens, inhibit protozoa, stimulate propionate production, and use alternative H_2_ sinks. The conditions for using natural compounds are that they should be safe for use in animals and humans, be effective in the long-term with different raw feedstuffs, have a low cost to reduce emissions from ruminants, and increase productivity in the livestock system. The initial research process with the evaluation of the phytochemical or natural compounds started with the in vitro screening and the second step consisted of experiments in vivo. Results of the in vitro experiments with dietary additives have shown to be inconsistent between experiments [26] and discrepancies exist when evaluated directly under in vivo conditions. However, the use of chitosan as a feed additive has demonstrated its potential in productive performance, nutrient utilization efficiency, increased protein, lactose, and unsaturated fatty acid concentration in the milk of cows under in vivo conditions [27,28,29].

## 2. Chitosan

Chitosan is a linear polysaccharide composed of two repeated units, D-glucosamine and N-acetyl-D-glucosamine, linked by β-(1→4)-linkages, characterized in terms of intrinsic properties such as molecular weight, viscosity, and degree of deacetylation (Figure 1). Chitosan is a collective name for a group of the partially or fully deacetylated biopolymer chitin, it is a natural compound, non-toxic, biocompatible, biodegradable, bioactive, muco-adhesive, and has been identified as safe for use in food in Japan (1983), Korea (1995), and the United States (Food and Drug Administration; 2012). It is a high molecular weight poly-cationic polymer, the second most abundant polysaccharide in nature, and is present in the structural exoskeleton of insects, crustaceans, mollusks, cell walls of fungi, and certain algae, but largely obtained from marine crustaceans [30]. Several gigatons of crustacean shell are produced annually and the extraction of chitin (10^6^–10^7^ tons), chitosan, and protein from this waste has added value [31,32]. It has antimicrobial properties against bacteria, filamentous fungi, and yeast, and even has virus, anti-inflammatory, antitumor activity, antioxidative activity, anticholesterolemic, hemostatic, and analgesic effects [33,34]. The application of chitosan either alone or blended with other natural polymers can be done in several ways such as silage inoculants, food processing, food preservation, textile, biotechnology, water treatment, pharmaceutical, tissue engineering, and the cosmetics industry [35,36]. Recent research in animal nutrition has focused on its potential to modulate rumen fermentation in beef or dairy cattle [37,38,39,40,41] and nutrient digestibility in cattle. The chitosan extraction process can be carried out in a chemical or biological way. The chemical method at an industrial scale starts with demineralization to eliminate the calcium carbonate and calcium chloride; deproteinization; decolorization (mainly astaxanthin and β-carotene); and finally alkaline deacetylation using sodium or potassium hydroxide [35,42,43]. The biological way, which is considered environmentally safe, uses lactic acid for demineralization, deproteinization by proteases, decoloration with acetone or organic solvents, and finally deacetylation by bacteria. In recent years, new extraction methodologies have been developed with the use of microwave irradiation [43]. The quality of the final product depends upon the raw material (crustaceans species), process of extraction, and seasonal variations [36,44].

## 3. Antimicrobial Mechanism

Recent research emphasizes the search for natural products with antimicrobial activity, low price, and high availability to reduce the use of chemicals and avoid drug resistance [45,46]. Chitosan has a broad spectrum of activity against different fungi, suppressing sporulation and spore germination, Gram-positive and Gram-negative bacteria, but lower toxicity toward mammalian cells (Figure 2). The antimicrobial mechanism of chitosan is complex and has not been fully described, however, the proposed mechanisms include interactions at the cell surface and outer membrane through electrostatic interactions or divalent cations, the replacement of Mg^2^ and Ca^2^ ions, the destabilization of cell membrane and leakage of intracellular substances, and the death of cells [28,47]. Other mechanisms that have been suggested are its chelating capacity in acid or neutral conditions for various metal ions including Ni^2^, Zn^2^, Co^2^, Fe^2^, Mg^2^, and Cu^2^ [48,49], and the inhibition of mRNA and protein synthesis in cell nuclei [50]. Bacteria appear to be generally less sensitive to the antimicrobial action of chitosan than fungi, regardless of the type of Gram or bacterial species [34]. The antimicrobial capacity involves intrinsic factors like molecular weight (depending on bacterial strains), hydrophilicity, crystallinity, solubility, and degree of deacetylation of the parent chitosan. Extrinsic factors include pH pka (6.3–6.5), ionic strength in the medium [51], temperature, and storage [36,52]. 

## 4. Effects of Chitosan in in Vitro Experiments

Chitosan has shown effects on feed intake, digestion, fermentation, and enteric methane production, however, the results generally disagree between the in vitro and in vivo studies (Table 1; Figure 3). In the in vitro tests, Belanche [39] found that chitosan changed rumen fermentation pattern and increased propionate production, decreasing cellulolytic bacteria such as *Fibrobacter*, *Butyrivibrio*, and *Ruminococcus*, hemicellulolytic bacteria such as *Eubacterium*, and increased amylolytic bacteria. This led us to speculate that the electrostatic interaction of chitosan and the destabilization of the cell membrane inhibited methanogens or metabolic pathways of methane synthesis and reduced methane production by 10 to 42%, increasing propionic acid and lactate as the fermentation products as a result of the use of chito-oligosaccharides as carbon sources by rumen microbiota [53]. That report (Belanche) [39] disagrees with the results of Goiri [53,54,55]**,** who tested different types of chitosan (with respect to the degree of deacetylation and molecular weight) at doses that did not affect the total volatile fatty acid production, but decreased true organic matter digestibility. These results could be related to its antimicrobial effect, since chitosan (85% degree of deacetylation, 200 mPas viscosity) reduced methane and maintained the volume of methane produced during incubation time. The previous results agree with those reported by Puspita [56]. Goiri [55] showed that chitosan was very effective in inhibiting biohydrogenation in vitro by increasing C18:1 t11 and conjugated linoleic acid (CLA) proportions regardless of fatty acids in the diet. These results can be related to the interaction with negatively charged free fatty acids, and supports the contention that chitosan alters rumen protozoa population, which agrees with the reports by Wencelová [57]**,** who found that chitosan did not affect fatty acid profile, linoleic acid, and trans-vaccenic acid, and decreased dry matter digestibility and total gas production, and showed a slight effect on methane production. 

## 5. Effects of Chitosan in In Vivo Experiments

### 5.1. Dry Matter Intake

Most of the reports showed that chitosan does not affect dry matter intake [29,38,41,57,58,59]. However, Dias [28] reported that in grazing beef steers supplemented with chitosan (0, 400, 800, 1200, or 1600 mg/kg DM), the increment recorded in dry matter intake (DMI) was probably associated with the increase in crude protein (CP) and neutral detergent fiber (NDF) digestibility. On the other hand, Rodrigues [37]**,** using Jersey heifers fed chitosan at 2.0 g/kg DM, observed a reduction in DMI, increased DM, CP, and NDF digestibilities, and reduced methane synthesis by an improved feed efficiency when chitosan was included as a feed additive.

### 5.2. Rumen Fermentation Pattern

Chitosan modified the rumen fermentation pattern by increasing propionic acid and decreasing the acetate:propionate ratio [28,29,59,60]. Increased propionate production could be explained by the reduction of Gram-positive bacteria [37]. Changes in molar proportions of VFA in the rumen when chitosan is used as a feed additive is conducive toward an improvement in the efficiency of utilization of metabolizable energy for growth (*k*_f_) [61]**,** which may lead to better animal performance and reduction in methane synthesis [62]. Thus, chitosan seems to work in the rumen much in the way as when grain (starch) is incorporated in the ration, by shifting the pattern of fermentation to a more propionic acid type of fermentation. Supplementation of chitosan improves feed efficiency of lactating cows, and increases the concentration of unsaturated fatty acids and cis-9,trans-11 CLA (rumenic acid; 18:2) [63] in milk. Del Valle [27] also reported an increase in nitrogen and energetic efficiency and reduced urinary nitrogen excretion [37]. Nitrogen utilization can be related to the reduction of the deamination rate of amino acids in the rumen and their absorption in the duodenum, which results in an overall improvement in the efficiency of N utilization. Mingoti [52] worked with dairy cows in mid-lactation and found that 100 and 150 mg of chitosan per kg body weight improved digestibility of crude protein, probably related to the proteolytic processes by ruminal bacteria or due to altered fermentation, although the mechanism remains unclear. Chitosan has no effect on rumen pH [28,29,38]. 

### 5.3. Rumen Microbial Population

The information presented has shown that chitosan exerts greater bactericidal effects against Gram-positive rather than on Gram-negative bacteria [64]. This antimicrobial action is enhanced at low pH values [64]. According to Zanferari [63]**,** chitosan in dairy cows fed without lipid supplementation decreased bacterial species such as the *Butyrivibrio* group and *B. proteoclasticus* related to the rumen biohydrogenation and reduced milk yield, although it increased the concentration in milk of unsaturated fatty acids (UFA) and cis-9,trans-11 CLA. Moreover, supplementing chitosan and soybean oil resulted in an antagonism that affected productive performance [63]. There is a lack of information in relation to rumen microbial population and their modifications with the use of different concentrations of chitosan. More work is required in this field since it could lead to improvements in our understanding of the effects of this feed additive.

### 5.4. Enteric Methane Emissions

The number of experiments to quantify methane emissions when chitosan is added to a ration is limited (Table 1). Henry [41]**,** working with the SF_6_ technique, reported that inclusion levels of chitosan of 0.5 and 1.0% of DM showed no effect on enteric CH_4_ production in cattle. They recorded differences between the fed, high concentrate diet since heifers produced almost 2.5 times less CH_4_ than the low concentrate ration [65].

## 6. Conclusions

In conclusion, the present review shows the lack of agreement between the experiments carried out to date, resulting in insufficient conclusive information on the methane mitigation potential of chitosan. However, the experiments performed show improved animal performance and nutrient utilization efficiency, increased propionate production, a reduced acetate:propionate ratio, and increased unsaturated fatty acid concentration in milk. Chitosan could be considered as a promising natural and abundant agent for enteric methane mitigation. The information reviewed suggests that chitosan can be used as a modulator of the fermentation pattern in the rumen, but future work should be aimed at exploring, under in vivo conditions, the synergistic or antagonistic effects with other feeds or nutrients such as lipids, pH effect, and the effect in different ruminant species as well as clarify the optimal doses, elucidate the antimicrobial mode of action at a molecular level, and quantify methane yield in experiments with greater accuracy such as with the respiration chamber or sulfur hexafluoride techniques.

## Figures and Tables

**Figure 1 animals-09-00942-f001:**
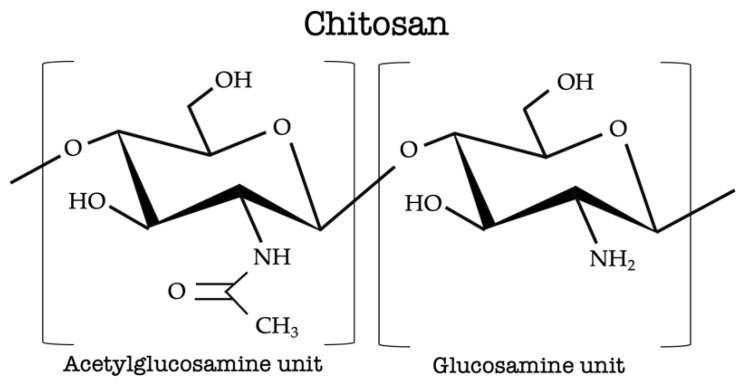
Chemical structure of chitosan.

**Figure 2 animals-09-00942-f002:**
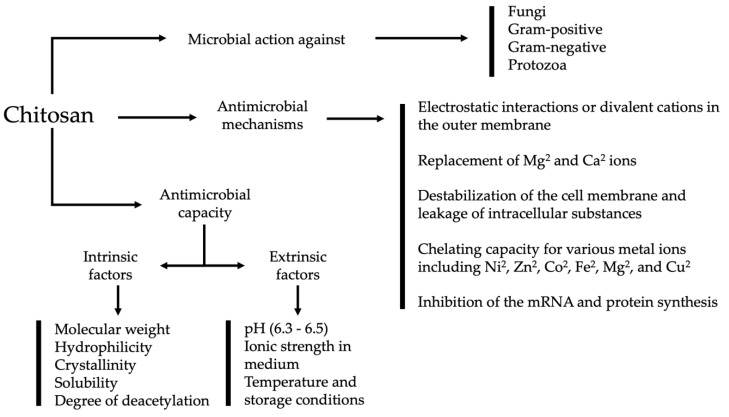
Antimicrobial mechanisms of chitosan.

**Figure 3 animals-09-00942-f003:**
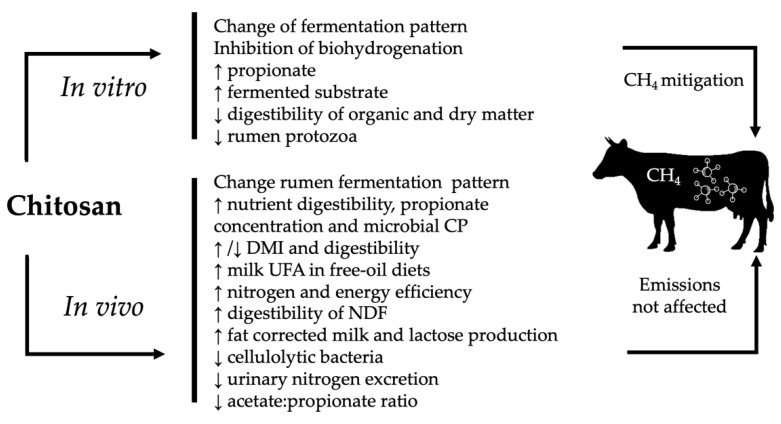
Mechanisms of action of chitosan described in the in vivo and in vitro experiments.

**Table 1 animals-09-00942-t001:** Effects of chitosan on rumen fermentation and methane emission**.**

Chitosan	Length of Experiment	Dosage	Substrate/Feed	Methane Determination	Results	Reference
>85% deacetylated with a viscosity equal to 140 mPas in 1% acetic acid solution at 25 °C	In vitro (18 days)	0 and 50 g/L of culture fluid	Forage-to-concentrate ratio 50:50	Gas chromatography	42% of reduction methane versus control, without modification of the rumen microbiota and VFA	[39]
Six different types	In vitro (24 and 144 h)	750 mg/L of culture fluid	Maize silage	Stoichiometry	Modification of rumen microbial fermentation and reduced 10 to 30% of methane	[53]
Three different types	In vitro (24 h)	0, 325, 750, and 1500 mg/L of culture fluid	Alfalfa hay and concentrate ratio 80:20; 50:50; 20:80	Stoichiometry	Effects were related to the nature of the feed and the characteristics of the additive, inconsistent results in methane reduction	[54]
Chitin and chitosan from Black Soldier Fly	In vitro (24 h)	10 and 20 g/L of culture fluid	Grass *Setaria splendida*: concentrate ratio 60:40	Gas chromatography	Methane production was not reduced and digestibility of OM and DM were decreased	[56]
Deacetylated chitin, poly (D-glucosamine) Sigma-Aldrich Co., St. Louis, MO, USA	In vitro (24 h)	100 mg/L of culture fluid	Meadow hay, barley grain, maize silage	Gas chromatography	Chitosan had an effect on IVDMD, total gas, slight effect on methane production, and some rumen ciliate genera	[57]
Deacetylation degree >95%; viscosity < 500 mPa s	In vitro (11 days)	750 mg/d of culture fluid	Grass hay and a concentrate mixture 10:90 using sunflower or rapeseed meal	Not quantified	Chitosan inhibited biohydrogenation	[55]
Deacetylation degree > 95%, viscosity < 500 mPa s	In vivo, in vitro Sheep (45 days)	0 and 136 mg/kg of BW	Alfalfa hay and concentrate at 50:50	Stoichiometry	Chitosan reduce NDF apparent digestibility, ruminal NH_3_-N concentration and modulates ruminal and fecal fermentative activity	[59]
Degree of deacetylation > 92% apparent density 0.64 g/mL; total ash ≤ 2.0%; pH 7.0–9.0; viscosity < 200 cPs	In vivo Cattle (84 days)	0, 50, 100 and 150 mg/kg BW	Corn Silage-concentrate 60:40	Not quantified	Chitosan shifted rumen fermentation, improved nutrient digestibility and propionate concentrations	[38]
Deacetylation degree of 86.6%	In vivo Cattle (84 days)	0 and 4 g/kg of DM	Corn silage-to- concentrate ratio 50:50	Not quantified	Improved feed efficiency, increased milk UFA concentration	[27]
Deacetylation degree ≥ 85%, 0.32 g/mL density, pH 7.90, and viscosity < 200 cPs	In vivo Cattle (105 days)	0, 400, 800, 1200 or 1600 mg/kg DM	Grazing *Urochloa brizantha* and concentrate at 150 g/100 kg of LW	Not quantified	Chitosan increased DMI and digestibility, propionate concentration and microbial crude protein	[28]
Deacetylation degree of 86.3%; 0.33 g/mL of apparent density, pH = 7.9, viscosity < 200 cPs, 1.4% ash, and 88.3% of DM	In vivo Cattle (98 days)	0, 75, 150, 225 mg/kg BW	Corn silage to concentrate ratio 63:37	Not quantified	In dairy cattle works like a modulator of rumen fermentation, increasing milk yield, propionate and nitrogen utilization	[29]
Deacetylation degree of 95%; apparent density of 0.64 g mL^−1^, 20 g kg^−1^ of ash, 7.0–9.0 of pH, viscosity < 200 cPs.	In vivo Cattle *(*25 days each period)	0, 2.0 g/kg Chitosan (CH) of DM. Whole raw soybean (WRS) 163.0 g/kg DM; and CH + WRS	Corn silage to concentrate ratio 50:50	Not quantified	Chitosan improved nutrient digestion and decrease DMI and reduce nitrogen excreted in feces	[37]
Deacetylation degree 90%	In vivo (21 days each period) and in vitro (24 h)	0.0, 0.5, and 1.0% of DM	High-concentrate (85%) Low concentrate (36%)	Sulfur hexafluoride (SF_6_)	In vivo: No effect on enteric methane emissions. In vitro: Low concentrate substrate increased methane production	[41]
Deacetylation degree of 86.6%; 0.33 g/mL of apparent density, pH of 8.81	In vivo Cattle (84 days)	50, 100 and 150 mg/kg BW	Corn silage to concentrate ratio 50:50	Not quantified	Improved nutrient digestibility without altering productive performance of dairy cows	[58]
Deacetylation degree 95%; viscosity < 200 cPs density 0.64 g/mL; pH 7.0–9.0	In vivo Cattle (84 days)	150 mg/kg BW	Maize silage: concentrate ratio 50:50	Not quantified	Chitosan increase the digestibility and reduce acetate to propionate relation	[60]
Deacetylation degree of 86.3%; apparent density of 0.32 g/mL, pH 7.9, viscosity of 50 cP at 20 °C	In vivo Cattle (92 days)	0 or 4 g/kg Chitosan (CH) or Whole Raw Soybean (WRS) of DM	Corn silage: concentrate ratio 50:50	Not quantified	CH + WRS affected ruminal fermentation, increased milk content of UFA, decreases nutrient intake, digestibility, microbial protein synthesis, and milk yield. CH in diets with no lipid supplementation improves feed efficiency of lactating cows	[63]

BW: Body weight; DM: Dry matter; OM: Organic matter; DMI: Dry matter intake; IVDMD: in vitro dry matter digestibility; UFA: Unsaturated fatty acids.

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
