# Peer review of "The Role of Chitosan as a Possible Agent for Enteric Methane Mitigation in Ruminants"

_animals, 2019, doi:10.3390/ani9110942_

Round 1

Reviewer 1 Report

General – I think you have done a good job of reviewing the topic, so most of my comments are editorial in nature.

many times you use the phrase ‘Experiments carried out to date’, or something like that. I believe that is intuitive (you can’t report on future experiments), so I suggest rephrasing to eliminate that redundancy.

Line                 Comment

19-20  “ …  decreasing … are global commitments,…. “  or perhaps “ Effort to increase …. and decrease …. Is a global commitment.  Either way, use ‘increase’ and ‘decrease’ , or ‘raise’ and ‘lower’

21 (VFA)   the apostrophe s is unnecessary

22  rephrase to clarify that this is an enteric loss, before absorption and metabolism

45 “…a marine product. Like other compounds…”

46 “… of extraction and storage. In addition, it is important…”

50  “ … production systems. Chitosan is a promising…”

51 “…  mitigating effects, but further research…”

54  Keywords should include ‘ruminant’ and ‘chitosan’

58  change ‘forcer’ to ‘effector’

60 “…livestock systems, particularly those that include ruminants {3}.”

68 – 71  I suggest adding a few words here to clarify that  feeding grains (low fiber diets) causes more H2 captured as propionate versus lost as methane, whereas high forage diets cause less capture as propionate and more lost as methane; methanogens and high-fiber diets go together

77 “….inconsistent {15}, with variation attributed to “

81 “ … production of methane {3,10.18}; therefore, in the last decade…”

84  Start a new paragraph with “Plants produce….”

88 Delete “that shows”

90 “ A range of feed additives including antibiotics has achieved …”

101  When you say ‘laboratory’, do you mean in vitro?  If so, say so.

102- 104  This statement needs citation and some clarification; what is the difference between ‘production parameters’ and ‘milk quality’? Do you mean feed efficiency?   I suggest replacing ‘improve’ with change;  milk quality is a nebulous descriptor.

122-124 “Recent research in animal nutrition has focused in its potential to modulate rumen fermentation in beef or dairy cattle {34-37} and nutrient digestibility in cattle {38}.”

125  “way at industrial scale..”

128 “way, considered environmentally safe, uses lactic acid…”

155-220  use present tense or past tense verbs consistently in these sections, e.g. ‘is’ or ‘was’,  ‘gives’ or ‘gave’,  ‘results; or ‘resulted’ ‘shows’ or showed’,  “modifies’ or ‘modified’.  I think past tense works better.

Table 1.   In the dosage column, do your readers a favor and avoid use of percentages; just convert 5 to appropriate units,  eg. mg/L

Author Response

Revisor 1.-

General – I think you have done a good job of reviewing the topic, so most of my comments are editorial in nature.

many times you use the phrase ‘Experiments carried out to date’, or something like that. I believe that is intuitive (you can’t report on future experiments), so I suggest rephrasing to eliminate that redundancy. Corrected

Line                 Comment

19-20  “ …  decreasing … are global commitments,…. “  or perhaps “ Effort to increase …. and decrease …. Is a global commitment.  Either way, use ‘increase’ and ‘decrease’ , or ‘raise’ and ‘lower’ (Increasing the efficiency of production systems and decreasing its environmental burden is a global commitment)

21 (VFA)   the apostrophe s is unnecessary Corrected

22  rephrase to clarify that this is an enteric loss, before absorption and metabolism “thus methane mitigation is a strategy to reach those goals, by rechanneling metabolic hydrogen (H2) into volatile fatty acids (VFA) to reduce the loss of energy as methane in the rumen which ranges from 2% (grain rations) to 12% (poor-quality forage rations) of gross energy intake”.

45 “…a marine product. Like other compounds…” Corrected

46 “… of extraction and storage. In addition, it is important…” Corrected

50  “ … production systems. Chitosan is a promising…” Corrected

51 “…  mitigating effects, but further research…” Corrected

54  Keywords should include ‘ruminant’ and ‘chitosan’ Corrected

58  change ‘forcer’ to ‘effector’  We revised and the most common word used is forcer than effector

60 “…livestock systems, particularly those that include ruminants {3}.” Corrected

68 – 71  I suggest adding a few words here to clarify that  feeding grains (low fiber diets) causes more H2 captured as propionate versus lost as methane, whereas high forage diets cause less capture as propionate and more lost as methane; methanogens and high-fiber diets go together Corrected 

Ration and type of feed affect the availability of metabolic hydrogen for CH4 synthesis in the rumen, feeding concentrate and grains (low fiber diets) reduce the population of methanogens (range 107 to 109 g-1) while more propionic acid is synthesized which utilize H2, whereas pasture-fed (methanogens 109 to 1010 g-1) ruminants yield higher concentrations of acetic acid in rumen liquor resulting in a higher availability of metabolic H2

77 “….inconsistent {15}, with variation attributed to “  Corrected

81 “ … production of methane {3,10.18}; therefore, in the last decade…” Corrected

84  Start a new paragraph with “Plants produce….” Corrected

88 Delete “that shows” Corrected

90 “ A range of feed additives including antibiotics has achieved …” Corrected

101  When you say ‘laboratory’, do you mean in vitro?  If so, say so. Corrected (in vitro)

102- 104  This statement needs citation and some clarification; what is the difference between ‘production parameters’ and ‘milk quality’? Do you mean feed efficiency?   I suggest replacing ‘improve’ with change;  milk quality is a nebulous descriptor. Corrected

122-124 “Recent research in animal nutrition has focused in its potential to modulate rumen fermentation in beef or dairy cattle {34-37} and nutrient digestibility in cattle {38}.” Corrected

125  “way at industrial scale..” Corrected

128 “way, considered environmentally safe, uses lactic acid…” Corrected

155-220  use present tense or past tense verbs consistently in these sections, e.g. ‘is’ or ‘was’,  ‘gives’ or ‘gave’,  ‘results; or ‘resulted’ ‘shows’ or showed’,  “modifies’ or ‘modified’.  I think past tense works better. Corrected

Table 1.   In the dosage column, do your readers a favor and avoid use of percentages; just convert 5 to appropriate units,  eg. mg/L Corrected

Reviewer 2 Report

Regarding the opening sentence -Ruminants are not a major cause of green house gas emissions. Plant fiber degradation is. This will happen if there are ruminants or not. Ruminants can capture some of what is released in the process.  Get this fact right.

Methane mitigation occurs with introducing starch or sugar into the ruminant diet, why wasn't chitosan impact compared to what occurs when grain is introduced ?

Regarding lines 155-165-I'm not sure the explanation is very clear or correct. Anything that shifts vfa production to more proprionate and less acetate will reduce methane production.

Lines 187-188--what do you mean that there is no change in vfa production, after you had just written that there was a shift to more proprionate production?

Author Response

Revisor 2.-

Regarding the opening sentence -Ruminants are not a major cause of green house gas emissions. Plant fiber degradation is. This will happen if there are ruminants or not. Ruminants can capture some of what is released in the process.  Get this fact right. Corrected

Ruminant husbandry is one the largest contributors to greenhouse gas emissions from the agriculture sector, particularly of methane gas, which is a byproduct of anaerobic fermentation of structural and non-structural carbohydrates in the rumen.

Methane mitigation occurs with introducing starch or sugar into the ruminant diet, why wasn't chitosan impact compared to what occurs when grain is introduced ?

Lines 198-200 Thus, chitosan seems to work in the rumen much in the way as when grain (starch) is incorporated in the ration, by shifting the pattern of fermentation to a more propionic acid type of fermentation. 

Regarding lines 155-165-I'm not sure the explanation is very clear or correct. Anything that shifts vfa production to more proprionate and less acetate will reduce methane production.

Lines 162 – 172 - Corrected

 Chitosan has shown effects on feed intake, digestion, fermentation, and enteric methane production, however, results generally disagree between in vitro and in vivo studies (Table 1; Figure 3). In in vitro tests, it was found, Belanche [39] that chitosan changed rumen fermentation pattern and increased propionate production, decreasing cellulolytic bacteria, such as Fibrobacter, Butyrivibrio and Ruminococcus, and hemicellulolytic bacteria such as Eubacterium and increased amylolytic bacteria. This led us to speculate that the electrostatic interaction of chitosan and the destabilization of the cell membrane inhibited methanogens or metabolic pathways of methane synthesis and reduced methane production by 10 to 42 %, increasing propionic acid, and lactate as fermentation products as a result of the use of chito-oligosaccharides as carbon sources by rumen microbiota [53].

Lines 187-188--what do you mean that there is no change in vfa production, after you had just written that there was a shift to more proprionate production? Corrected

Lines 193 – 194 Chitosan modified the rumen fermentation pattern by increasing propionic acid and decreasing the acetate:propionate ratio [28,29,59,60].